# The Effect of Two Types of Back Pillow Support on Transversus Abdominis and Internal Oblique Muscle Fatigue, Patient Satisfaction, and Discomfort Score during Prolonged Sitting

**DOI:** 10.3390/ijerph20043742

**Published:** 2023-02-20

**Authors:** Rungthip Puntumetakul, Thiwaphon Chatprem, Pongsatorn Saiklang, Arisa Leungbootnak

**Affiliations:** 1Department of Physical Therapy, Faculty of Associated Medical Sciences, Khon Kaen University, Khon Kaen 40002, Thailand; 2Research Center in Back, Neck, Other Joint Pain and Human Performance, Khon Kaen University, Khon Kaen 40002, Thailand; 3Division of Physical Therapy, Faculty of Physical Therapy, Srinakharinwiroj University, Nakhon Nayok 26120, Thailand; 4Human Movement Sciences, Department of Physical Therapy, Faculty of Associated Medical Sciences, Khon Kaen University, Khon Kaen 40002, Thailand

**Keywords:** rubber back pillow, foam back pillow, patients’ satisfaction, discomfort score, deep trunk muscle fatigue

## Abstract

Natural rubber is considered an economic plant in Thailand and is used to manufacture many products. Foam back pillows have proven to have various benefits for the lower back. However, no study has compared the effects of foam and rubber pillows. Therefore, the current study aimed to compare the efficacy of foam and rubber pillows on transversus abdominis and internal oblique muscle fatigue, patient satisfaction, and discomfort scores during 60 min of prolonged sitting. Thirty healthy participants were invited to the study and randomized into three sitting conditions over three consecutive days. The three groups were as follows: control, foam pillow, and rubber pillow. Our results revealed that the discomfort score increased with the sitting time in all three groups (*p* < 0.05). The control group had the highest discomfort when compared to the rubber pillow group at 30 min (T4; *p* = 0.007) and 60 min (T7; *p* = 0.0001), as well as the foam pillow group at 60 min (T7; *p* = 0.0001). Participants were more satisfied sitting with the two types of back pillows at the initial time (T1; *p* = 0.0001) and at 60 min (T7; *p* = 0.0001) when compared with the control group. Furthermore, the participants were more satisfied with using rubber pillows rather than foam pillows throughout the sitting period (*p* = 0.0001). The control group experienced more transversus abdominis and internal oblique muscle fatigue at 60 min (T7) of sitting compared to the initial time (T1) (*p* = 0.038). Thus, sitting with pillow support can decrease deep trunk muscle fatigue, and using a pillow made from natural rubber may ensure greater satisfaction and less discomfort for the user.

## 1. Introduction

Sedentary workers experience increased levels of inactivity, with a high proportion of prolonged sitting (≥30 min) [1,2]. Sedentary workers in Thailand reported recurring low back pain (LBP) and 63% reported that their LBP was aggravated by sitting during working hours [3]. Sitting for an extended period is considered to heighten the risk of LBP [4,5]. Furthermore, LBP may cause socioeconomic burdens such as prolonged loss of function, decreased work productivity, and increased medical costs [6,7]. 

Deep trunk muscles contribute to spinal stability [8] and are separated into superficial muscles or deep muscles [9,10]. Deep muscles compose the transverse abdominis (TrA), internal oblique (IO), and lumbar multifidus (LM). Furthermore, deep muscles are shorter in length, attached directly to the vertebrae, and primarily responsible for generating sufficient force for segmental spinal stability [9,10]. 

The TrA muscle initially arises from the iliac crest, the lower six ribs, and the middle and lateral raphe of the thoracolumbar fascia, and it passes medially to insert at the linea alba [11]. The TrA acts like a corset for the lumbar spine, tightens through the thoracolumbar fascia, and collaborates with the inferior IO muscle fibers to produce intra-abdominal pressure (IAP) during contraction [12]. The TrA and IO have been recognized as crucial in spinal unloading [13]. Fredericson and Moore (2005) reported that the TrA and IO are the primary stabilizers of the spine [14]. In healthy people, the TrA and IO have also been shown to contract 30 milliseconds before shoulder movement and 110 milliseconds before leg movement, and this is theoretically done to stabilize the lumbar spine [15,16]. 

Previous studies have reported that the continuous contraction of trunk muscles in prolonged seated postures could cause deep trunk muscle fatigue, and this is particularly true for the TrA and IO muscles [17]. Furthermore, prolonged sitting reduces the intervertebral disks’ ability to act as shock-absorbing hydraulic cushions [18,19,20]. Reductions in disk height could increase compressive stress on sensitive spinal structures [21,22] and may stimulate nociceptor activity, leading to pain [22]. Increasing deep trunk muscle fatigue can influence lumbar stability and ultimately lead to LBP [17,23,24]. 

According to McKenzie’s concept, LBP may stem from hypo-lordosis of the lumbar spine [25]. Therefore, appropriate curves are essential for reducing and preventing LBP symptoms. Active lumbar extension exercises have been shown in numerous trials to reduce LBP when sitting [13,22]. However, individuals may find it challenging to fit this exercise within their working hours. 

Many studies have focused on supporting lumbar lordotic curves during prolonged sitting to relieve pain and maintain mobility in the lumbar spine region [26,27]. Prommanon et al. (2014) compared two interventions, and the results showed that back pillows (foam material) are more effective than physical therapy in reducing pain and enhancing lumbar range of motion [26]. Kompayak et al. (2016) also reported that back pillows (foam material) could superiorly reduce pain intensity, enhance quality of life, increase lumbar range of motion, reduce functional impairment, and increase patient satisfaction when compared to lumbar support in people with chronic LBP [27]. However, the back pillows in previous studies were made from foam, whereas other materials, such as rubber, may convey certain advantages, including enhanced softness, increased flexibility, and prolonged usage time; rubber is also considered an economic plant of Thailand [28]. 

To our knowledge, no studies have compared the immediate effects that back pillow supports made from foam and rubber materials have on the TrA and IO muscles during prolonged sitting. Therefore, the current study aimed to investigate the immediate effects of back pillow support (foam and rubber materials) on fatigue in the TrA and IO muscles, discomfort scores, and participant satisfaction during prolonged sitting.

## 2. Materials and Methods

### 2.1. Ethics Statement 

This randomized crossover study was conducted at the Research Center in the Back, Neck, Other Joint Pain, and Human Performance (BNOJPH) laboratory at Khon Kaen University. The Khon Kaen University Ethics Committee (HE 632261, Khon Kaen, Thailand, 17 December 2020) approved the current study. 

### 2.2. Study Population Recruitment 

Between January 2021 and April 2021, 30 healthy participants were recruited via social media advertisements. The inclusion criteria were as follows: (i) without low LBP for at least the previous six months [29], (ii) between 20 and 69 years old [30], and (iii) a sedentary lifestyle (sitting more than two hours per day) [13,17]. The exclusion criteria were as follows: (i) had a neurological deficit, (ii) had any joint arthritis or inflammation, (iii) had spine pathology, or (iv) was pregnant [17]. Based on Hertzog’s (2008) guidelines for a pilot study’s sample size, the current study required at least 30 participants [31]. 

### 2.3. Procedure 

A screening process was used to determine participants’ eligibility for the study, and demographic data were recorded through direct interviews. Thirty healthy participants were asked to sign informed consent forms before participating. Participants who met the inclusion criteria were then asked to visit the research laboratory on three consecutive days and were randomly selected to sit in three positions, as shown in Figure 1.

One researcher informed the participants throughout the study. Another researcher measured the outcomes, including trunk muscle fatigue, participant satisfaction, and discomfort scores. The assessment times for each outcome are shown in Figure 2. 

### 2.4. Outcome Measurement

#### 2.4.1. Discomfort Scores

For the assessment of discomfort scores, an 11-point numerical rating scale (NRS) was used. This scale was scored from 0 (without any discomfort) to 10 (extreme discomfort). The discomfort was scored three times, at T1, T4, and T7. Ferreira-Valente et al. (2011) claimed that this NRS can be used to determine specific indications [32], including discomfort differences between males and females [26,27,32]. In addition, this NRS is sensitive to variations in discomfort scores among chronic LBP patients [33]. The NRS used has excellent reliability and validity [32] and describes a technique that can be tested for both physical and cognitive disability, even in the elderly [34].

#### 2.4.2. Patients’ Satisfaction

The global perceived effect (GPE) scale was selected to assess the participants’ satisfaction. The GPE is a 10-point scale ranging from −5 (worst symptom) to 0 (no change) to +5 (close to normal improvement). The intraclass correlation coefficient (ICC) of the GPE was between 0.90 and 0.99, which was assessed as excellent [26,27,35].

#### 2.4.3. Trunk Muscle Fatigue

The skin of each participant was cleaned with an alcohol pad before the researcher attached one pair of surface electromyography (sEMG) electrodes (EL 503), with an electrical contact surface area of 1 cm^2^ and a center-to-center spacing of 2.5 cm, on the dominant limb site of the TrA and IO muscles [36]. The locations for the attached sEMGs on the TrA and IO muscles were inferior to the anterior superior iliac spine.

The electromyography data were recorded at 2000 samples per second using the Wireless Bipolar Cometa Mini Wave Plus 16-channel EMG system (Cometa, Bareggio, Italy), with an online band-pass filter (10–500 Hz) and a 60 Hz notch filter (power line in Thailand). For the experimental task, participants performed the required sitting condition for 60 min (i) in a control condition (sitting without support), (ii) with a foam pillow (sitting with a back pillow made of foam material), and (iii) with a rubber pillow (sitting with a back pillow made of rubber material) in a random order on three consecutive days (Figure 1). The EMG data of the TrA and IO muscles were recorded seven times (T1–T7), as shown in Figure 2, and each time was captured for one minute. 

The raw EMG signals were full-wave rectified and represented as median frequency (MF) values. EMG normalization was the method by which the magnitude of muscle activation was expressed as a percentage of the muscle’s activity during a calibrated test condition. The current study evaluated the maximum voluntary isometric contraction (MVIC) of the trunk muscle utilizing the methods outlined by Imai et al. (2010) for normalizing data [37].

To determine the MVIC values in the TrA and IO muscles, three muscle tests were performed. A rest period of two minutes was allowed between the tests to avoid muscle fatigue [38,39,40]. The participants performed the tests in the supine position [36,37]. All the normalized MF values achieved during each test were expressed as a percentage of MVIC (%MVIC). 

### 2.5. Statistical Analysis 

All analyses were performed using SPSS version 19.0 software (SPSS Inc., Chicago, IL, USA). Mean and standard deviation (SD) were used to assess the participants’ demographics. A Shapiro–Wilk test was performed to check the data distribution. 

A paired *t*-test was used to compare patient satisfaction within the groups. The difference in trunk muscle fatigue (T1–T7) and discomfort scores (measured at T1, T4, and T7) within groups for nonnormally distributed data were analyzed using a Friedman test, and a Wilcoxon signed-rank test was used for post-hoc analysis. The differences in trunk muscle fatigue, discomfort scores, and patient satisfaction between groups were analyzed using a Wilcoxon signed-rank test. The significance level was *p* < 0.05. 

## 3. Results

The participants’ demographic characteristics are shown in Table 1. Of the 30 participants, 17 (56.67%) were female. The mean age of all participants was 40.97 ± 13.77 years, and all participants had a normal body mass index (18.1 ± 2.14).

### 3.1. Discomfort Score

Three groups of participants who sat for 60 min were evaluated for their back comfort using a discomfort score. As shown in Figure 2, the back discomfort score was measured three times (at T1, T4, and T7), and the outcomes are displayed in Table 2.

As shown in Table 2, when compared within groups, increased sitting time caused participants to experience a statistically significant increase in back discomfort in all experimental groups, particularly in the control group, who sat without a back pillow and had the highest discomfort score. When comparing the groups, it was found that the control group had significantly greater back discomfort than the rubber pillow group. The control group experienced greater back discomfort than the foam pillow group after 30 min and 60 min of sitting. The group using the foam pillow reported statistically significantly greater back discomfort than the group using the rubber pillow.

Participants experienced the most back discomfort while sitting without a back pillow at all time points. Furthermore, when participants used the rubber back pillow, they experienced the least back discomfort compared to the other groups, at a *p*-value of 0.0001.

### 3.2. Participants’ Satisfaction

The participants’ satisfaction was measured twice while they sat for 60 min (as shown in Figure 2), and the results are displayed in Table 3.

When comparing the patients’ satisfaction within groups, the satisfaction of the control group decreased significantly after 60 min of sitting. However, within the rubber group, there was a statistically significant increase in satisfaction when participants sat for 60 min.

Compared between groups, participants were more satisfied with back pillows (both types) than without back support at T1. Furthermore, at T1, it was revealed that the satisfaction of participants using rubber back pillows was statistically considerably higher than that of the participants using foam pillows.

The effect of satisfaction at T1 was consistent with that observed at T7, in that satisfaction was greater in the back pillow group than in the control group. Participants using rubber pillows reported significantly higher satisfaction levels than those using foam pillows.

### 3.3. Trunk Muscle Fatigue

Table 4 shows the participants’ trunk muscular fatigue while sitting for 60 min. This research focuses on the TrA and IO muscles. TrA and IO muscle fatigue were observed only in the control group at T7. In addition, there was no difference between the groups.

## 4. Discussion 

The current study investigated the effect of using rubber pillows compared to foam pillows on TrA and IO muscle fatigue, discomfort scores, and participant satisfaction. Thirty healthy participants were asked to sit for 60 min on three consecutive days under three sitting conditions. The three conditions were (i) a control group (sitting without support), (ii) a foam pillow group (sitting with a foam pillow), and (iii) a rubber pillow group (sitting with a rubber pillow). 

Patients’ discomfort scores increased in all groups while sitting for only 30 min, particularly in the control group, where participants sat without a back pillow and had the highest discomfort level compared with the other two pillow groups. Previous studies that showed prolonged sitting can induce discomfort, even in healthy individuals, supported these results [41,42]. 

After 30 and 60 min of sitting, the control group had more back discomfort than the foam and rubber pillow groups. The discomfort score was lower in the pillow group, and this finding was concordant with the study of Prommanon et al. (2015), which found decreased pain intensity in participants who received back pillows in addition to physical therapy [26]. Furthermore, Kompayak et al. (2016) demonstrated that using back pillows in addition to physical therapy was more effective at decreasing pain intensity than using lumbar support in addition to physical therapy [27]. 

Using a back pillow may have psychological effects and cause participants to maintain correct posture and good ergonomics [43], leading to minimized excessive movements of the spine and relieving impact on the lumbar joint, thereby straightening the spinal column and decreasing pressure within the spine [44]. Additionally, back pillows are designed to have a lumbar spinal curvature similar to that of normal individuals, which may result in the maintenance of lumbar spinal curvature. This can prevent irregularities from increasing tension in the structure of the lumbar spine, thus minimizing the risk of pain [45].

The foam pillow group had much higher back discomfort than the back rubber pillow group. This may be because the natural rubber pillow had higher resilience, which made participants feel better when using it. Thus, rubber is typically used in mattress and pillow production [46]. 

Regarding participant satisfaction, the current study discovered that participants who sat with a back support device were more satisfied than those without one at the initiation of sitting (T1) or after 60 min of sitting (T7). Participants were more satisfied with the rubber back pillow than with the foam pillow at both T1 and T7. Kompayak et al. (2016) compared patient satisfaction between the foam pillow and lumbar support, and they reported that participants who used the foam pillow achieved higher satisfaction than lumbar support users [27]. However, the current study compared two types of pillow materials—foam and rubber—that have never been studied. The current study’s results show that the natural rubber pillow achieved higher user satisfaction than the foam pillow, which may be due to its increased flexibility and elasticity, which can better fit lumbar curves [47].

The current study reported that participants without a back pillow (control group) had TrA and IO fatigue at T7 (sitting for 60 min). Sitting for prolonged periods may cause TrA and IO muscle fatigue due to the continuous contraction of the TrA and IO muscles in seated postures [17,20]. The lumbar multifidus is passively stretched during prolonged sitting, resulting in the TrA and IO muscles increasing their co-contraction activity to balance back muscle forces. Consequently, the TrA and IO muscles become fatigued over time [17,48]. A deficiency in the activation of the TrA and IO muscles reduces muscular support to the spine, causing impairment of motor coordination and increased stress on spinal structures [49]. Therefore, the control group experienced significantly increased discomfort during prolonged sitting.

However, the foam and rubber groups showed no difference in MDF values over 60 min of sitting. The TrA and IO muscles play a crucial stabilizing role in the lumbopelvic region and reduce stress on spinal structures [17,49]. Furthermore, the pillow groups induced a lumbar lordotic curve while sitting. Thus, these pillows can reduce the flexed posture associated with disk compression [22,50] and prolonged contraction of the TrA and IO muscles [17,48]. The findings from our study suggest that these pillows might be appropriate while sitting to prevent TrA and IO muscle fatigue in individuals who usually spend an extended period sitting. However, while the current study referenced the sample size guidelines for pilot studies, this may be a limitation. Thus, further studies should be conducted with larger sample sizes to confirm the results. 

## 5. Conclusions 

The current study was the first to compare the effects of foam and rubber pillows on TrA and IO muscle fatigue, discomfort scores, and patient satisfaction. The authors recommend that individuals who sit for prolonged periods during the day use back support to delay deep trunk muscle fatigue and reduce discomfort scores. Furthermore, rubber pillows may provide enhanced comfort due to their softness and flexibility compared to foam pillows.

## Figures and Tables

**Figure 1 ijerph-20-03742-f001:**
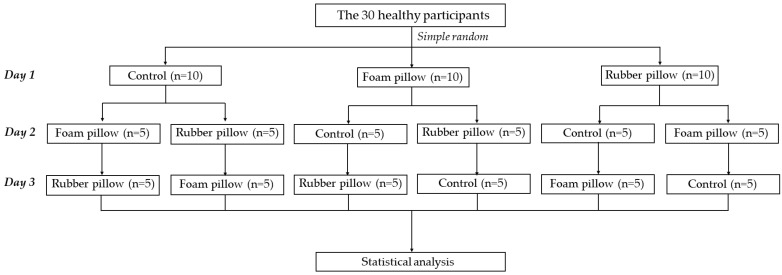
Overview of the study.

**Figure 2 ijerph-20-03742-f002:**
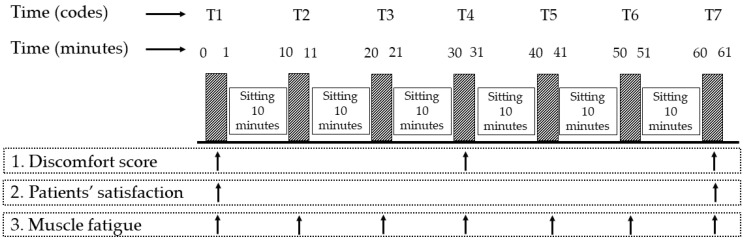
The assessment times for each outcome.

**Table 1 ijerph-20-03742-t001:** General characteristics of 30 healthy participants.

Variables	Frequency (%)	Mean ± SD
**Age** (years)		40.97 ± 13.77
**Gender**, n (%) ▪ Male ▪ Female	13 (44.33) 17 (56.67)	
**BMI** (kilogram/meter^2^)		18.1 ± 2.14
**Education level**		
▪ High school ▪ Bachelor’s degree ▪ Master’s degree	14 (46.67) 8 (26.67) 8 (26.67)	
**Smoking** ▪ Yes ▪ No	6 (20) 24 (80)	
**Underlying disease** ▪ Yes ▪ No	8 (26.68) 22 (73.33)	
**Exercise status** ▪ Yes ▪ No	16 (53.33) 14 (46.67)	

Abbreviations: SD, standard deviation; BMI, Body Mass Index.

**Table 2 ijerph-20-03742-t002:** Comparison of discomfort scores in the back region within the group and between the experimental groups.

Groups	Discomfort Score			*p*-Value within Group
	T1	T4	T7	
**Control**	0 (0–0) ^a,b,^*	1 (0–3) ^c,^*	3 (1.75–6)	0.0001 *
**Foam pillow**	0 (0–0) ^d,e,^*	0 (0–2) ^f,^*	2 (0–4.25) ^k,m,^*	0.0001 *
**Rubber pillow**	0 (0–0) ^g,h,^*	0 (0–1) ^i,j,^*	0 (0–2.25) ^l,^*	0.0001 *
***p*-value between group**	0.223	0.007 *	0.0001 *	

Note: Discomfort score reported as median (interquartile range); *p*-value from Wilcoxon signed-rank test. * *p*-value < 0.05; ^a^ statistically different from T4; ^b^ statistically different from T7; ^c^ statistically different from T7; ^d^ statistically different from T4; ^e^ statistically different from T7; ^f^ statistically different from T7; ^g^ statistically different from T4; ^h^ statistically different from T7; ^i^ statistically different from T7; ^j^ statistically different from the control group; ^k^ statistically different from the control group; ^l^ statistical different from the control group; ^m^ statistical different from the rubber pillow group.

**Table 3 ijerph-20-03742-t003:** The comparison of patients’ satisfaction within and between groups.

Groups	Patients’ Satisfaction		*p*-Value within Group
	T1	T7	
**Control**	0 (0–0) ^a^*^b^*	(−2) (−4–0) ^d^*^e^*	0.002 *
**Foam pillow**	1 (0–3.25) ^c^*	2 (0–4) ^f^*	0.389
**Rubber pillow**	2.5 (0–4)	3 (1.75–5)	0.006 *
***p*-value between group**	0.0001 *	0.0001 *	

Note: Patient’s satisfaction reported as median (interquartile range); *p*-value from Wilcoxon signed-rank test. * *p*-value < 0.05; ^a^ statistically different from foam pillow; ^b^ statistically different from rubber pillow; ^c^ statistically different from rubber pillow; ^d^ statistically different from foam pillow; ^e^ statistically different from rubber pillow; ^f^ statistically different from rubber pillow.

**Table 4 ijerph-20-03742-t004:** Trunk muscle (TrA and IO) fatigue while sitting for 60 min.

Muscle	Groups	Time							*p*-Value within Group
		T1	T2	T3	T4	T5	T6	T7	
**TrA & IO (Hz)**	**Control**	97.61 ^a^* (74.69–127.00)	94.5 (69.92–119.28)	93.2 (68.57–112.46)	91.91 (70.98–142.62)	88.53 (65.52–130.34)	87 (64.22–130.34)	80.8 (67.07–122.65)	**0.038 ***
	**Foam pillow**	84.5 (72.96–130.99)	87 (64.16–123.45)	90.35 (74.88–123.54)	92.44 (54.44–119.05)	93.71 (70.33–131.51)	87.82 (70.27–121.81)	81.56 (70.56–127.78)	0.392
	**Rubber pillow**	106.32 (82.85–142.25)	97.7 (84.11–132.65)	89.36 (77.80–111.17)	84.49 (69.87–130.75)	86.96 (77.13–131.17)	84.08 (71.73–128.94)	82.15 (64.65–128.13)	0.088
	***p*-value between group**	0.136	0.67	0.648	0.733	0.587	0.648	0.421	

Note: Muscle fatigue was reported as median (interquartile range); *p*-value from Wilcoxon signed-rank test. * *p*-value < 0.05; ^a^ statistically different from T7.

## Data Availability

The data will be available for anyone who wishes to access them for any purpose and contract should be made via the corresponding author (thiwch@kku.ac.th).

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
