# Peer review of "The Effect of Two Types of Back Pillow Support on Transversus Abdominis and Internal Oblique Muscle Fatigue, Patient Satisfaction, and Discomfort Score during Prolonged Sitting"

_ijerph, 2023, doi:10.3390/ijerph20043742_

Round 1
Reviewer 1 Report
This is a very interesting study, but several issues need to be resolved before publication. Please see my comments attached.

Author Response
We sincerely appreciate your consideration of our work. We have carefully responded to answer all of your concerns. The revised version of the manuscript is marked with a yellow highlight changed for reviewer 1 as in the attached document.

Reviewer 2 Report
Review of a manuscript -Manuscript ID: ijerph-2161237
The paper presents interesting research relevant to the prevention of back pain syndromes.
The title encourages you to read the content of the article.
The summary is complete and meets the requirements.
The purpose of the publication was clearly defined.
In the introduction section, I don't see an explanation of the abbreviation LBP. Although the abbreviation is widely known, it would be useful to put it in brackets when first used and explain it, as well as the presentation of the abbreviations: " transverse abdominis (TrA), internal oblique (IO), and lumbar multifidus (LM)."
The research results confirm the assumptions of the work.
References - please standardize the record in accordance with the requirements of the journal.
The work needs correction.
Author Response
We appreciate your consideration of our work. The revised version of the manuscript is marked with a green highlight changed for reviewer 2 as attached document.

Round 2
Reviewer 1 Report
I am happy with the changes!
Regards